# Proximal clustering between BK and Ca$_V$1.3 channels promotes functional coupling and BK channel activation at low voltage

Oscar Vivas[1,2]*, Claudia M Moreno[1,2], Luis F Santana[2†], Bertil Hille[1†]

[1]Department of Physiology and Biophysics, University of Washington, Seattle, United States; [2]Department of Physiology and Membrane Biology, University of California, Davis, Davis, United States

**Abstract** Ca$_V$-channel dependent activation of BK channels is critical for feedback control of both calcium influx and cell excitability. Here we addressed the functional and spatial interaction between BK and Ca$_V$1.3 channels, unique Ca$_V$1 channels that activate at low voltages. We found that when BK and Ca$_V$1.3 channels were co-expressed in the same cell, BK channels started activating near −50 mV, ~30 mV more negative than for activation of co-expressed BK and high-voltage activated Ca$_V$2.2 channels. In addition, single-molecule localization microscopy revealed striking clusters of Ca$_V$1.3 channels surrounding clusters of BK channels and forming a multi-channel complex both in a heterologous system and in rat hippocampal and sympathetic neurons. We propose that this spatial arrangement allows tight tracking between local BK channel activation and the gating of Ca$_V$1.3 channels at quite negative membrane potentials, facilitating the regulation of neuronal excitability at voltages close to the threshold to fire action potentials.

*For correspondence: ovivas@ucdavis.edu

†These authors contributed equally to this work

Competing interests: The authors declare that no competing interests exist.

## Introduction

This paper characterizes coupling between BK (K$_{Ca}$1.1, mSlo1, KCNMA1) and Ca$_V$1.3 (α1D, CACNA1D) channels. We present evidence for multi-channel complexes that these two channels form. In nerve and muscle, depolarization of the membrane activates voltage-gated calcium channels (Ca$_V$) leading to calcium influx. This increase in calcium induces the activation of BK channels. The activation of BK channels serves as negative feedback, tending to repolarize the membrane and to close Ca$_V$ channels. Therefore, understanding the functional and physical relation between BK and Ca$_V$ channels is critical to elucidate the biophysical and physiological actions of this feedback mechanism.

BK channels are expressed in nearly every excitable cell and their activation depends on both voltage and intracellular calcium (*Cui et al., 2009*). Without calcium ([Ca$^{2+}$]$_i$ = 0), BK channels activate only at very depolarized voltages with a V$_{1/2}$ >100 mV (*Yan and Aldrich, 2010*). The affinity of BK channels for calcium ions ranges between 1 and 10 µM (*Contreras et al., 2013*). Therefore, a rise in calcium concentration from ~100 nM resting levels to the micromolar range is needed to increase the open-channel probability of BK channels at −50 to 0 mV (*Barrett et al., 1982*).

Opening of a single calcium channel raises local calcium to more than 100 µM within a few tens of nanometers from the inner mouth of the channel, but most of these ions are quickly buffered–within microseconds (*Simon and Llinás, 1985*; *Roberts, 1994*). BK channels have been found in close proximity to all subfamilies of Ca$_V$ channels, suggesting a molecular interaction (*Roberts et al., 1990*; *Roberts, 1994*; *Marrion and Tavalin, 1998*; *Berkefeld and Fakler, 2008*; *Müller et al.,*

*2010*; *Rehak et al., 2013*). This association can be recapitulated in heterologous systems and has been shown to reconstitute functional nanodomains (*Berkefeld et al., 2006*). The physical arrangement and channel stoichiometry of these putative complexes in the nanodomain are still not clear. Some models depict a 1:1 relation between $Ca_V$ and BK channels arranged at distances as close as 10 nm (*Fakler and Adelman, 2008*; *Cox, 2014*). Other models are similarly effective in reproducing the functional coupling without a 1:1 stoichiometry (*Prakriya and Lingle, 2000*; *Montefusco et al., 2017*). Thus, a comprehensive characterization of $Ca_V$-BK interactions within functional nanodomains remains to be completed.

The activation threshold of BK channels tends to reflect the activation threshold of associated $Ca_V$ partners. Dihydropyridine-sensitive, L-type $Ca_V1.3$ channels are unique in their activation profile; they can start activating at voltages as negative as −65 mV (*Platzer et al., 2000*; *Lipscombe et al., 2004*). These channels play essential roles in the auditory system (*Koschak et al., 2001*; *Michna et al., 2003*; *Brandt et al., 2003*, *2005*), in the pacemaker cells of the heart (*Mangoni et al., 2003*; *Christel et al., 2012*; *Torrente et al., 2016*), in chromaffin cells (*Marcantoni et al., 2010*), and in neurons (*Putzier et al., 2009*; *Liu et al., 2014*; *Cooper et al., 2015*; *Lu et al., 2015*; *Stanika et al., 2016*). Deficiency of $Ca_V1.3$ channels leads to deafness and bradycardia (*Platzer et al., 2000*).

Despite the potential relevance for many excitable cells, functional and structural coupling between BK and $Ca_V1.3$ channels is poorly defined. Here, we tested the hypothesis that coupling between these channels can activate BK channels at very negative voltages. This would promote a negative feedback mechanism whereby BK-channel opening would oppose further opening of $Ca_V1.3$ channels even before the membrane potential reaches the firing threshold for action potentials. In order to activate BK channels at these low voltages, the $Ca_V1.3$ and BK channels must be organized in close proximity (<20 nm), so that nanodomains of calcium forming near the inner mouth of the calcium channel overlap with the calcium sensors of the BK channel.

## Results

To test the concept of spatial proximity, we (i) compared the activation properties of BK channels when co-expressed with two different $Ca_V$ isoforms, (ii) used classical biophysical and biochemical approaches to test for coupling and direct interaction, and (iii) imaged BK and $Ca_V$ channels with super-resolution microscopy to characterize the spatial distribution of these channels.

### Expression of $Ca_V1.3$ channels shifts the voltage dependence of activation of BK channels to more negative potentials

We tested the hypothesis that BK and $Ca_V1.3$ channels are functionally coupled. A prediction of this hypothesis is that the voltage dependence of activation of the BK channels should follow the activation curve of the $Ca_V1.3$ channels (*Berkefeld et al., 2006*). Therefore, we compared activation of BK channels in a cell line (tsA-201 cells) co-expressing BK with either low-voltage activated $Ca_V1.3$ channels or high-voltage activated $Ca_V2.2$ channels. These cells do not express BK or any of the isoforms of $Ca_V$ channels endogenously (*Berjukow et al., 1996*; *Zhu et al., 1998*; *Avila et al., 2004*).

As expected, the inward calcium currents from cells expressing only $Ca_V2.2$ or $Ca_V1.3$ (*Figure 1A*) are blocked by cadmium, a general blocker of $Ca_V$ channels, but they differ in their gating kinetics. *Figure 1B* shows the voltage dependence of activation gating as conductance-voltage (G-V) curves fitted with Boltzmann functions. The midpoint of activation of the $Ca_V1.3$ channels is 35 mV more negative than that for the $Ca_V2.2$ channels. Outward $K^+$ currents from cells expressing only BK channels are insignificant in the voltage range that we used (*Figure 1—figure supplement 1*). When $Ca_V$ channels are co-expressed with the BK channels, the same voltage steps elicit an outward current that is blocked by cadmium (*Figure 1C and D*). These results confirm that calcium influx through $Ca_V$ isoforms is required for the activation of BK channels in the expression system. As anticipated, the mid-point voltage for activation of BK channels depends on the co-expressed $Ca_V$ channel isoform (*Figure 1E*). For a quantitative comparison, we use the fitted Boltzmann functions in *Figure 1* to interpolate the voltage for 5% activation of the channels. Loosely, we will call this the activation threshold, although activation is actually continuously graded at all voltages. In this terminology, the activation threshold for $Ca_V2.2$ channels was −13.8 mV and for the co-expressed BK channels it was −20.2 mV, whereas for $Ca_V1.3$ channels the activation threshold was −48.8 mV and for the co-

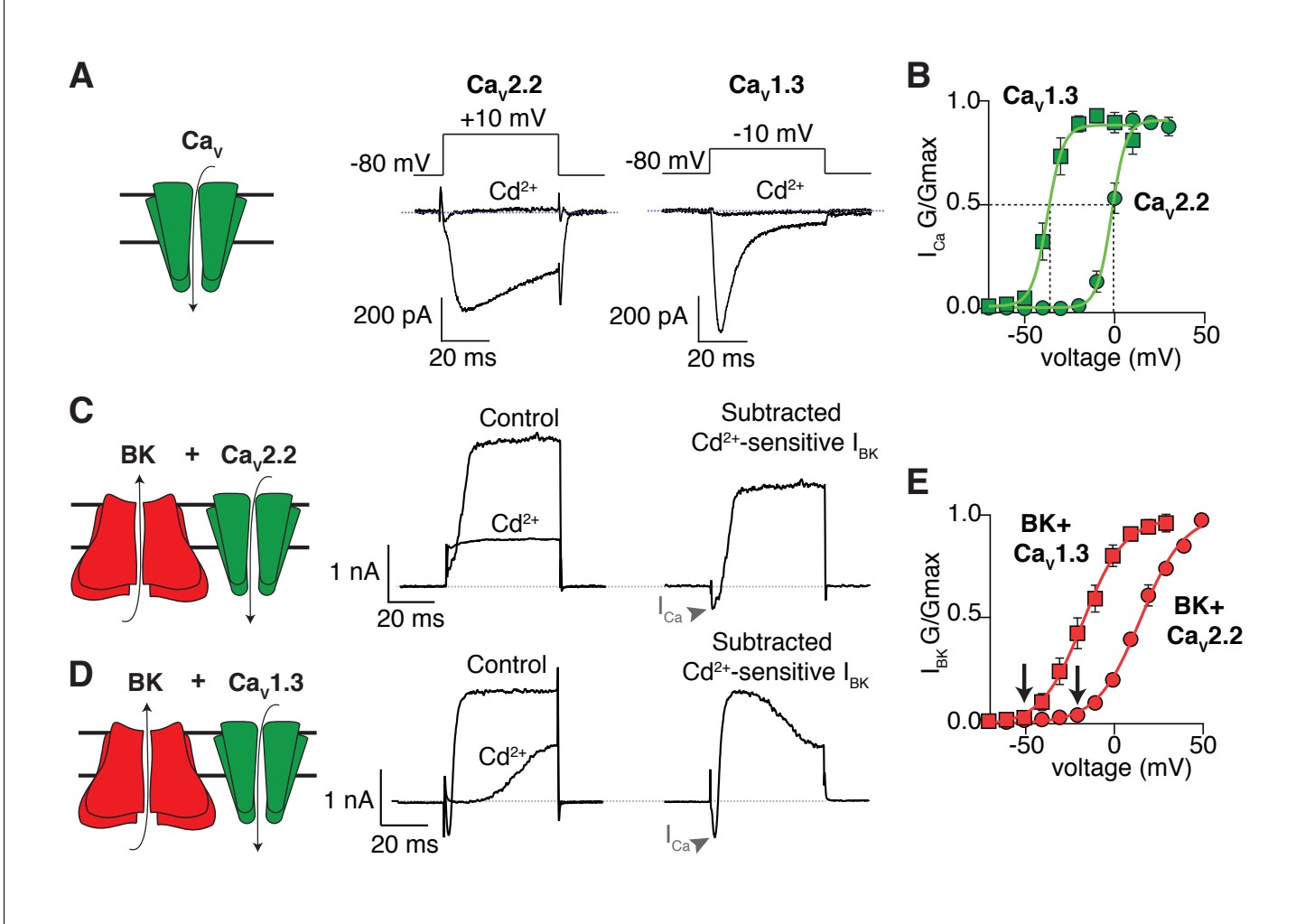

**Figure 1.** Shift in the activation profile of BK channels by co-expression with different $Ca_V$ channels. (A) Representative cadmium-sensitive inward (negative) calcium currents activated with the voltage protocol indicated above in tsA-201 cells transfected only with $Ca_V2.2$ or $Ca_V1.3$ channels. (B) Average conductance-voltage (G–V) relationships of $Ca_V2.2$ channels (green circles, n = 8) and $Ca_V1.3$ channels (green squares, n = 9) showing that $Ca_V1.3$ channels activate at more negative potentials than $Ca_V2.2$ channels. (C) Representative outward (positive) potassium currents activated with the same voltage protocol (above) in tsA-201 cells co-transfected with BK + $Ca_V2.2$ channels before and after application of cadmium. The subtracted current shows the inward calcium current (arrow heads) followed by the outward potassium current. (D) Same as in C but tsA-201 cells were co-transfected with BK + $Ca_V1.3$ channels. (E) Average G-V relationships for BK channels co-expressed with $Ca_V2.2$ (red circles, n = 9) or $Ca_V1.3$ channels (red squares, n = 7) showing that BK channels activate at more negative potentials when activated by $Ca_V1.3$ than when activated by $Ca_V2.2$. Arrows point at activation threshold. The smooth G-V curves in (B) and (E) are fitted Boltzmann functions: G/Gmax = 1/(1+ exp(-(V-$V_{mid}$)/slope)), where V is membrane voltage, $V_{mid}$ is the voltage for the half-maximal activation, and slope describes the voltage dependency of channel gating. For the curves in (B), the $Ca_V$ parameters $V_{mid}$ and slope are −37 mV and 4 mV for $Ca_V1.3$ and −2 mV and 4 mV for $Ca_V2.2$. For the curves in (E), the BK parameters are −17 mV and 12 mV with $Ca_V1.3$ and +16 mV and 12.3 mV with $Ca_V2.2$.

The following figure supplements are available for figure 1:

**Figure supplement 1.** BK currents in tsA-201 cells require co-expression of $Ca_V1.3$ channels.

**Figure supplement 2.** Activation of BK channels is slower when coupled with $Ca_V2.2$ channels.

expressed BK channels it was −52.3 mV. In short, there is quantitative agreement with the hypothesis that the $Ca_V$ channel with the more negative activation voltage allows BK channels to activate at correspondingly more negative potentials. The activation kinetics of BK channels has also been shown to change when coupled with different $Ca_V$ isoforms (**Berkefeld and Fakler, 2008**). In

agreement, the time constant of BK channel activation at 0 mV is shorter when co-expressed with $Ca_V1.3$ (8 ± 2 ms, n = 7) than when is co-expressed with $Ca_V2.2$ channels (18 ± 5, n = 9), resembling the difference in activation kinetics between $Ca_V1.3$ (0.7 ± 0.1 ms, n = 9) and $Ca_V2.2$ (4.5 ± 0.7 ms, n = 8) channels (*Figure 1—figure supplement 2*).

## $Ca_V1.3$ channels are in close proximity to BK channels

The activation of BK channels at −40 mV when co-expressed with $Ca_V1.3$ channels suggests that these two channels are in close proximity. We tested this hypothesis several ways. As a crude first approximation, we recorded from membrane patches on tsA-201 cells transfected with both BK and $Ca_V1.3$ channels with the hypothesis that the channels should be close enough to allow us to observe single-channel events. The pipette solution was similar to Ringer's solution except that the calcium concentration was increased from 2 to 20 mM. A 1 s voltage step from −80 mV to 0 mV was applied to activate $Ca_V1.3$ channels and to let calcium in only at the membrane patch. If nearby BK channels were present, they would then open. BK channels outside the membrane patch are not expected to open given the decay in calcium concentration. The solution in the bath, outside the membrane patch, contained high potassium to maintain the cytoplasmic potential close to 0 mV, helping to prevent channel openings outside the pipette. To prevent the activation of BK channels inside the patch by calcium coming from $Ca_V1.3$ channels outside the patch, 2 mM EGTA was added to the bath solution. Indeed, unitary outward $K^+$ currents with multiple levels were recorded from these patches (*Figure 2A*, *Figure 2—figure supplement 1* for raw trace), reflecting the activation of $Ca_V1.3$ channels and subsequent activation of BK channels. As a measure of BK channel activity, we calculated $NP_o$ the product of number of channels times probability of opening. $NP_o$ in these patches was 3.7 ± 0.7 (n = 8, *Figure 2B*). $NP_o$ decreased to 0.8 ± 0.4 (n = 7, p value = 0.002) after bath application of the dihydropyridine blocker nifedipine (10 µM) suggesting that $Ca_V1.3$ channels need to be activated to induce these outward current events. Openings were abolished and $NP_o$ decreased to 0.2 ± 0.1 (n = 7, p value = 0.0004) after the sequential application of a BK channel blocker, 500 nM paxilline, confirming the identity of BK channels coupled with $Ca_V1.3$ channels (*Figure 2B*).

$Ca_V1.3$ and BK channels have been previously shown to co-immunoprecipitate from brain extracts (*Grunnet and Kaufmann, 2004*), suggesting a direct interaction. However, expression of these channels in a heterologous system might not recapitulate a direct interaction, perhaps requiring additional proteins present uniquely in brain tissue. Thus, we investigated whether, as in neurons, BK and $Ca_V1.3$ channel are physically associated in tsA-201 cells using co-IP approaches. A comparison between protein extractions from tsA-201 cells transfected with both BK and $Ca_V1.3$ channels, only one channel (BK or $Ca_V1.3$), or none, shows that the BK antibody stains the input blot when BK channels are expressed alone or together with $Ca_V1.3$ channels (*Figure 2C*, left gel). The BK antibody also stains immunoprecipitates pulled down with $Ca_V1.3$ antibodies (n = 3, *Figure 2C*, right gel), suggesting that some level of interaction between these channels is reconstituted when expressed in a heterologous system and that no other proteins specific to brain tissue are required for this interaction.

A third biophysical approach tested whether BK channels are distributed within nanometer distances from $Ca_V1.3$ channels. We recorded potassium currents in whole-cell configuration and dialyzed calcium buffers with different binding rate constants into the cytoplasm (through the patch pipette) (*Naraghi and Neher, 1997*). The buffers limited calcium diffusion to specific distances from the calcium source. The equation of *Prakriya and Lingle, 2000*, predicts that dialysis of 10 mM EGTA restricts local elevations of intracellular calcium to a radius of 100 nm from the calcium source, whereas dialysis of 10 mM BAPTA restricts diffusion to a radius of 10 nm (*Figure 3A*). Tellingly, potassium currents activated by $Ca_V1.3$ channels were reduced by dialysis with BAPTA but not EGTA (*Figure 3B*). Interestingly, some current remains even with the dialysis of BAPTA, suggesting that some BK channels lie closer than 10 nm to $Ca_V1.3$ channels. In contrast, potassium currents were reduced by dialysis of either BAPTA or EGTA in tsA-201 cells co-expressing BK and $Ca_V2.2$ channels (*Figure 3C*), suggesting that BK channels are distributed farther than 100 nm from $Ca_V2.2$ channels. These observations imply that BK and $Ca_V1.3$ channels organize in close proximity, creating a functional unit with preferential co-localization in a range of less than 100 nm.

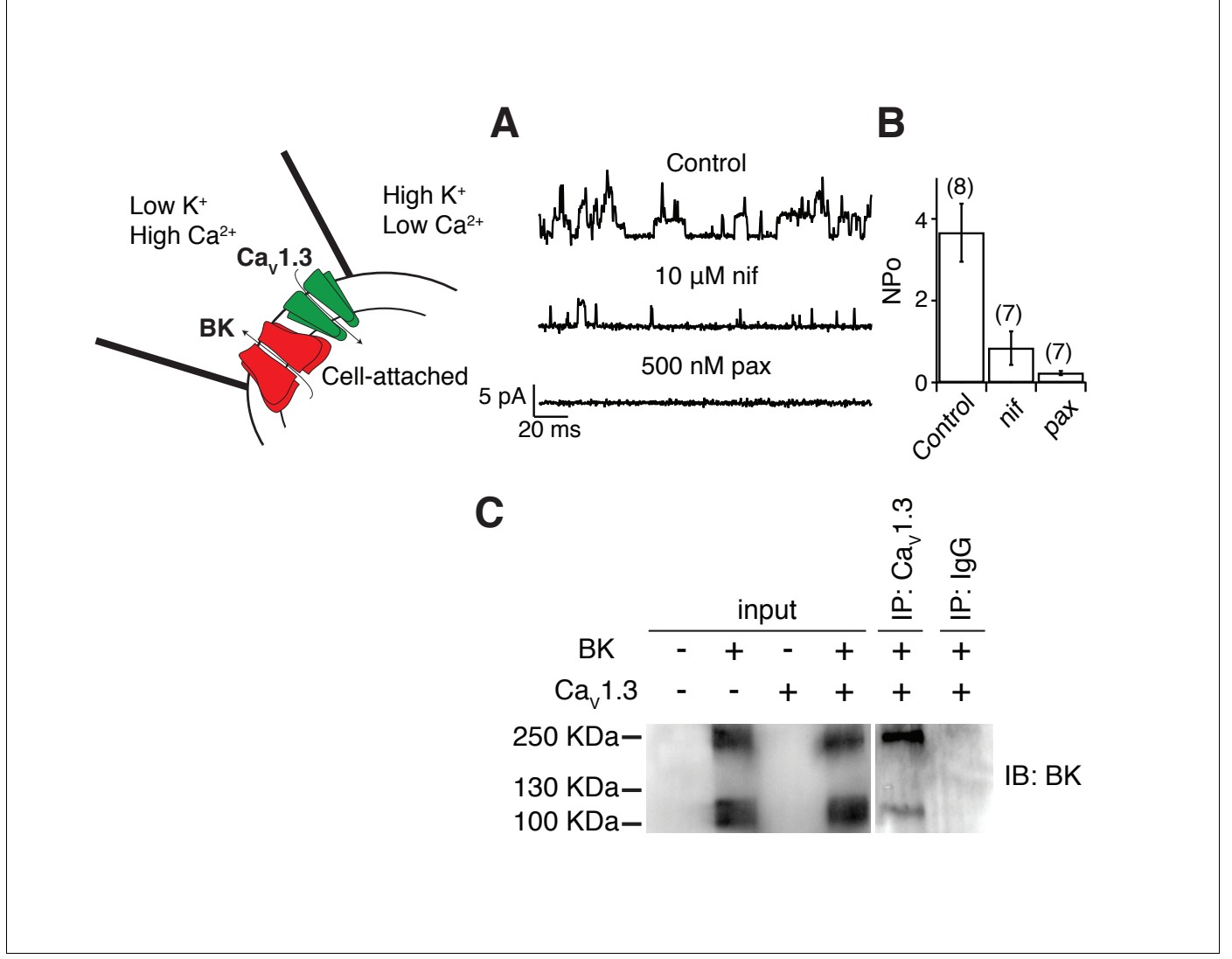

**Figure 2.** BK and Ca$_V$1.3 channels are close enough to be recorded in a small membrane patch and to be detected by co-immunoprecipitation. (**A**) Representative single-channel recordings from tsA-201 cells expressing BK and Ca$_V$1.3 channels. Recordings were done in cell-attached mode, and outward (positive) BK currents were elicited by 1 s voltage steps to 0 mV, sufficient to activate Ca$_V$ channels, from a holding potential of $-80$ mV. BK channel openings were decreased by sequential application of 10 μM nifedipine and 500 nM paxilline to the bath solution. The single-channel event-detection algorithm of pClamp 10.2 was used to measure single-channel opening amplitudes and to calculate the product of number of channels times probability of opening (NP$_o$). (**B**) Quantification of BK channel activity (NP$_o$) from each condition in **A**. (**C**) (left gel) Detection of BK channel protein by Western blot in tsA-201 cells transfected with BK cDNA only or with BK and Ca$_V$1.3 cDNA. BK channels were not detected when cells were not transfected or when only Ca$_V$1.3 cDNA was transfected. The molecular weight of the monomeric BK channel is around 130 KDa. The band located at around 250 KDa likely corresponds to dimers. (right gel) Detection of BK channels in a pulldown assay with Ca$_V$1.3 antibody (n = 3 gels from independent experiments). BK channels were not detected if proteins were pulled down with an unspecific IgG.
The following figure supplement is available for figure 2:

**Figure supplement 1.** Raw BK single-channel recording.

## Clusters of Ca$_V$1.3 channels surround BK channels

We proceeded to determine the spatial organization of BK and Ca$_V$1.3 channels with nanometer resolution. Super-resolution microscopy can resolve subcellular structures with a localization precision of approximately 20 nm that varies with the fluorophore used (*Dempsey et al., 2011*). We first checked how well our microscope could resolve previously studied structures. *Figure 4—figure*

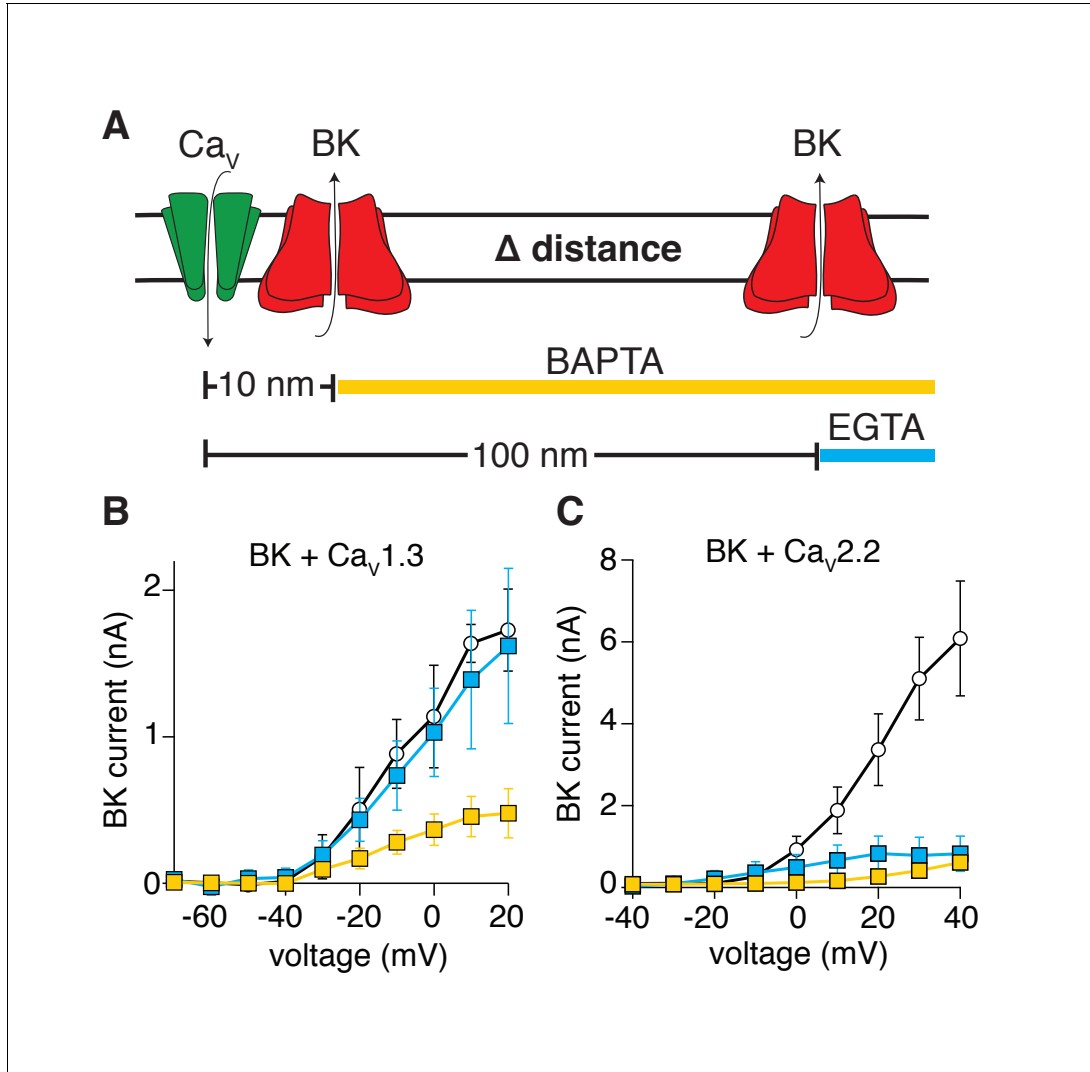

**Figure 3.** Activation of BK channels by Ca$_V$1.3 channels is blocked by fast calcium buffers. (**A**) Diagram of the experimental strategy. BAPTA will buffer calcium farther than 10 nm from the calcium source. EGTA will buffer calcium farther than 100 nm from the calcium source. (**B**) Comparison of current-voltage relations for BK channels coexpressed with Ca$_V$1.3 channels and measured with either control pipette solution (empty circles, n = 5), EGTA-containing solution (blue squares, n = 5), or BAPTA-containing solution (yellow squares, n = 7). Recordings were done in whole-cell configuration. (**C**) Comparison of current-voltage relations for BK channels coexpressed with Ca$_V$2.2 channels and measured with either control pipette solution (empty circles, n = 7), EGTA-containing solution (blue squares, n = 9), or BAPTA-containing solution (yellow squares, n = 7).

*supplement 1* shows localization maps from tsA-201 cells stained for α-tubulin. In agreement with previous reports (*Burnette et al., 2011*), these single microtubules exhibit a full-width-at-half-maximum of 33 ± 1 nm. This result validates our experimental approach.

tsA-201 cells co-expressing BK and Ca$_V$1.3 channels were fixed and stained for super-resolution imaging. Secondary antibodies conjugated to Alexa-647 or Alexa-568, were used to detect single molecules in two-color microscopy in the same cell. To determine bleed-through between spectral channels, we obtained super-resolution images of microtubules in tsA-201 cells labeled with secondary Alexa-647 or 568. We followed the same imaging settings and imaging sequence normally used in all our experiments (imaging first the 647 channel and then the 568 channel). We calculated the fraction of bleed-through by dividing the total number of localization events detected in the 'wrong' channel by the total number of localization events detected in the 'correct' channel for both fluorophores (*Figure 4—figure supplement 2*). The bleed-through from the emission of Alexa-568 into the 647 channel was 0.092 ± 0.032 (n = 5) and the bleed-through from the emission of Alexa-647

into the 568 channel was 0.008 ± 0.001 (n = 3), suggesting that the bleed-through between these two fluorophores is minimal.

In *Figure 4A*, pixels positive for BK channels (red) formed multi-pixel aggregates, with a median area of 1600 nm$^2$ (n = 5 cells). They are much larger than the expected size for a single ion channel, which has known dimensions of 13 by 13 nm (*Wang and Sigworth, 2009*; *Tao et al., 2017*). Therefore, we interpreted these aggregates as clusters of multiple BK channels. In numerous cases, the clusters of BK channels (red) were surrounded by variable numbers of clusters of Ca$_V$1.3 channels (green) rather than showing any fixed stoichiometry or geometry (*Figure 4A*). Only 18 ± 5% of the BK clusters were not surrounded by Ca$_V$1.3 clusters within a radius of 200 nm (n = 5 cells).

To provide a more quantitative description of the super-resolution maps, we measured the distance between BK and Ca$_V$1.3 clusters from edge to edge. The frequency distribution for the distance between Ca$_V$1.3 and BK-positive pixels (*Figure 4C*, green bars) was empirically fitted by a bell-shaped curve with a peak at 20 nm, our resolution limit, and a half width of 60 nm. This distribution fits with the distances suggested by the use of calcium buffers (*Figure 3B*), where BK channels are closer than 100 nm and even BAPTA could not block all the current, meaning that some BK channels probably lie at distances less than 10 nm from Ca$_V$1.3 channels. In another approach to describe the distribution of Ca$_V$1.3 channels around BK channels, we defined concentric areas of 20 nm width around the irregular BK clusters and measured the percentage of area occupied by Ca$_V$1.3 positive pixels in each concentric area (*Figure 4D*, green squares). To avoid bias in the analysis, data were analyzed from the whole cell instead of selected regions of interest (ROIs). The first point corresponds to Ca$_V$1.3-positive pixels that actually overlap with BK clusters. This plot shows a skewed distribution of Ca$_V$1.3 channels towards areas within 50 nm from BK clusters, indicating again that Ca$_V$1.3 clusters are preferentially concentrated in close proximity to BK clusters. Importantly, only 17 ± 2% of Ca$_V$1.3 clusters are at <200 nm from BK channels while the other 83% of Ca$_V$1.3 clusters are not localized close to any BK cluster (n = 5 cells).

To rule out that this distribution was a fixation artifact, we imaged the clusters from live tsA-201 cells in an AiryScan microscope which doubles the resolution achieved with a diffraction-limited confocal microscope. For these experiments we expressed BK channels tagged with GFP and Ca$_V$1.3 channels tagged with mRuby. Similar to what we observed in fixed cells, live cells exhibited BK channel puncta surrounded by Ca$_V$1.3 channel puncta (*Figure 4—figure supplement 4*).

We repeated the super-resolution study in cells co-expressing BK channels with Ca$_V$2.2 channels. *Figure 4B* shows representative ROIs from these cells. The median BK cluster size is not significantly different when BK channels are co-expressed with Ca$_V$2.2 channels (2000 nm$^2$, n = 4 cells). In comparison to the results with Ca$_V$1.3 channels, there is much less evidence for an association of Ca$_V$2.2 with BK channels. The frequency distribution for the distance between Ca$_V$2.2 and BK-positive pixels is roughly described by a bell-shaped curve centered at 100 nm from BK-positive pixels (*Figure 4C*, cyan bars). This distribution also fits with the distances suggested by the use of calcium buffers in *Figure 3C*. In addition, Ca$_V$2.2-positive pixels are more randomly distributed around BK channels (*Figure 4D*, cyan circles). Altogether, these results suggest that Ca$_V$1.3 channels are more likely to be close to BK channels than are Ca$_V$2.2 channels in our heterologous expression system. These results do not contradict the observed coupling between BK and Ca$_V$2.2 channels in neurons (*Marrion and Tavalin, 1998*). Rather they suggest some differences in the coupling mechanism and its ability to reconstitute in an expression system.

In the super-resolution maps, there were 2.5 times more Ca$_V$1.3 clusters (1843 ± 587) with a median cluster size of 1600 nm$^2$ than Ca$_V$2.2 clusters (741 ± 273) with a median cluster size of 400 nm$^2$. Presumably a higher density of Ca$_V$1.3 channels is expressed in these cells. We confirmed this observation by comparing the calcium current densities. The peak inward current density was −38 ± 11 pA/pF (n = 11) in cells co-transfected with Ca$_V$1.3 and only −10 ± 3 pA/pF (n = 9) in cells co-transfected with Ca$_V$2.2 (p=0.04). Because a higher expression of Ca$_V$1.3 channels might explain the closer average proximity of Ca$_V$1.3 and BK channels, we compared the distribution of BK channels to a random distribution of Ca$_V$1.3-positive pixels. To generate the random distribution from our data while keeping the number, size, and shape of each cluster, we binarized the maps of Ca$_V$1.3-positive pixels and randomized them with a custom macro written in MATLAB (*Source code 1*). Then, we merged the original map of BK-positive pixels unchanged with the new random-distribution map of Ca$_V$1.3-positive pixels for each cell (*Figure 4—figure supplement 5*) and analyzed the distribution around BK clusters. The distribution curve of randomized Ca$_V$1.3 pixels did not

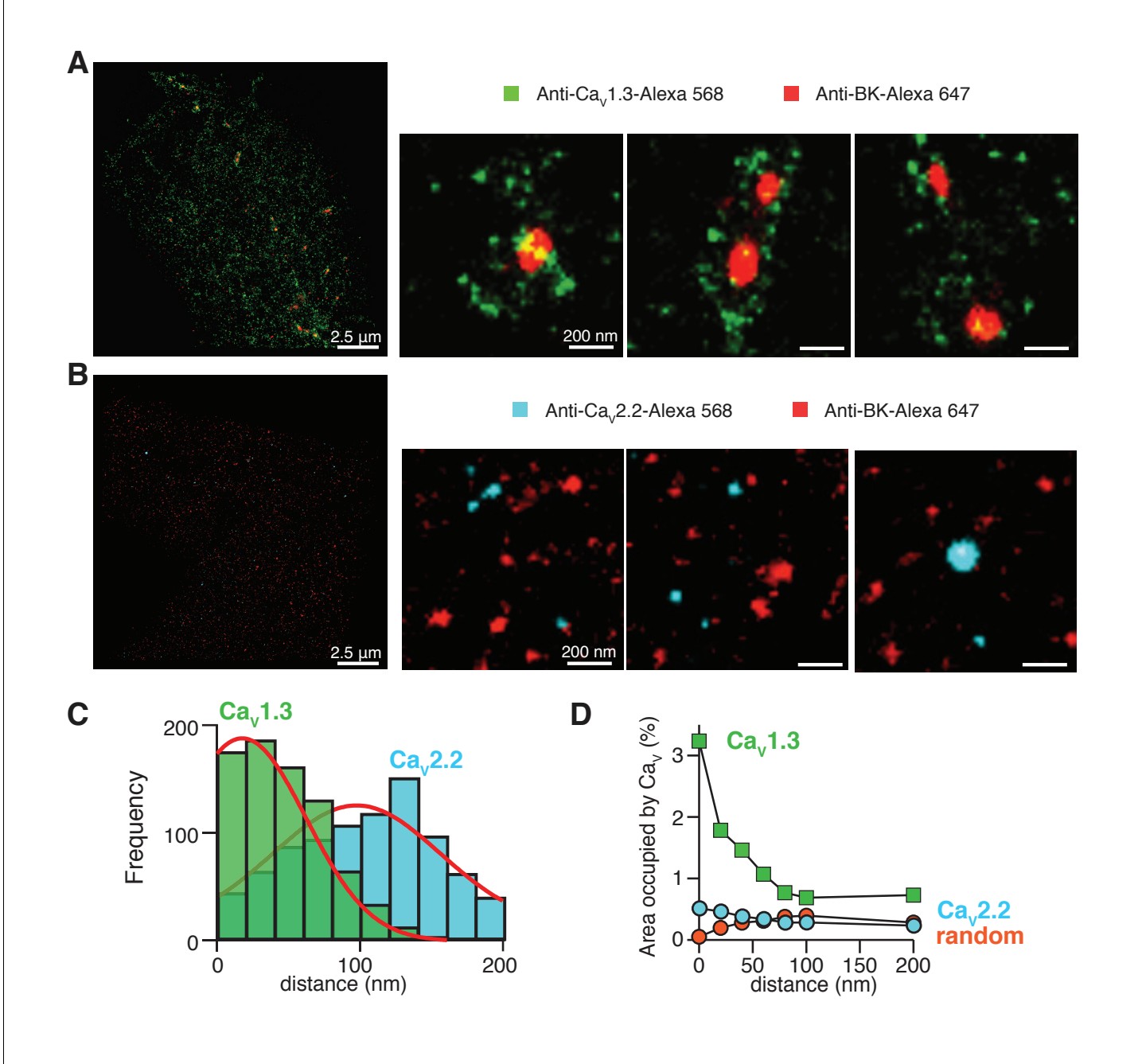

**Figure 4.** Ca$_V$1.3 channels are preferentially distributed in close proximity to BK channels. (**A**) Three representative single-molecule localization maps from cells co-expressing BK (red) and Ca$_V$1.3 (green) channels, showing individual clusters of BK channels surrounded by several clusters of Cav1.3 channels. (**B**) Three representative single-molecule localization maps from cells co-expressing BK (red) and Ca$_V$2.2 (cyan) channels, showing no particular association between BK and Ca$_V$2.2 channels. (**C**) Frequency distribution of the distance from BK-positive clusters to Ca$_V$1.3-positive (green, n = 5 cells) or Ca$_V$2.2-positive (cyan, n = 4 cells) clusters. (**D**) Distribution analysis based on the percentage of area occupied by Ca$_V$1.3 clusters (green squares, n = 5 cells), Ca$_V$2.2 clusters (cyan circles, n = 4 cells), or randomized Ca$_V$1.3 clusters (orange circles, n = 5 cells) as a function of the distance from BK channels.

The following figure supplements are available for figure 4:

**Figure supplement 1.** Single-molecule localization maps of α-tubulin in tsA-201 cells.

**Figure supplement 2.** Little bleed-through between wavelength channels in super-resolution imaging.

*Figure 4 continued on next page*

*Figure 4 continued*

**Figure supplement 3.** Control single-molecule localization maps of BK and Ca$_V$1.3 channels in tsA-201 cells.

**Figure supplement 4.** BK-Ca$_V$1.3 clustering is not an artifact of fixation.

**Figure supplement 5.** Generation and analysis of random single-molecule localization maps.

depend on the position of BK clusters (*Figure 4D*, orange circles), being more similar to the distribution observed for Ca$_V$2.2 channels. This result rejects the possibility that a higher expression level of Ca$_V$1.3 channels alone accounts for the greater coupling with BK channels.

## BK and Ca$_V$1.3 channels form clusters in sympathetic and hippocampal neurons

Does this clustered distribution require overexpression of channels? Hippocampal and sympathetic neurons abundantly express BK and Ca$_V$1.3 channels, where they are known to serve important physiological functions (*Storm, 1987*; *Lancaster and Nicoll, 1987*; *Gu et al., 2007*). Thus, we investigated the spatial distribution of BK and Ca$_V$1.3 channels in these neurons using super-resolution imaging of immunostained neurons. As neurons express both Ca$_V$1.3 and Ca$_V$1.2 channels, it is important to be certain of the specificity of the Ca$_V$1.3 antibody. The antibody used in this work recognizes a highly variable site within the intracellular loop between domains II and III with no homology to the Ca$_V$1.2 channel (*Figure 5—figure supplement 1A*) and its specificity has been tested previously (*Hell et al., 1993*). To further assess the antibody's specificity in our super-resolution experiments, we co-transfected tsA-201 cells with Ca$_V$1.3 and Ca$_V$1.2-EGFP channels and co-immunostained with anti-Ca$_V$1.3 and anti-GFP. *Figure 5—figure supplement 1B* shows that the two antibodies label different structures. In fact, the overlap between the two labels is only 2.2 ± 0.5% (n = 4), supporting that the antibody used against Ca$_V$1.3 channels does not recognize Ca$_V$1.2 channels.

Similar to the distribution observed in the heterologous expression system, BK channels formed clusters surrounded by clusters of Ca$_V$1.3 channels (*Figure 5A,B*). Specific secondary antibodies conjugated to Alexa-647 or Alexa-568 were used to detect BK and Ca$_V$1.3 channels, respectively. Reciprocal immunostaining using Alexa-647 for the Ca$_V$1.3 and Alexa-568 for the BK channels was also performed. The BK-Ca$_V$1.3 clustering organization was not dependent on the secondary antibody used (*Figure 5—figure supplement 2*). Independent of the secondary antibody used, all the cells were included in the distribution analysis. The median BK cluster size is 2800 nm$^2$ in SCG neurons (n = 7), significantly larger than the median BK cluster size in hippocampal neurons (2000 nm$^2$, n = 6) or the expression system (1600 nm$^2$, n = 5). The distribution analysis in both types of neurons also showed a skewed distribution of Ca$_V$1.3 clusters occupying areas adjacent to BK clusters (*Figure 5C, E*, green squares), indicating that Ca$_V$1.3 clusters are preferentially concentrated in close proximity to BK clusters, similar to the distribution observed in the heterologous expression system. Interestingly, whereas in hippocampal neurons 3% of the pixels inside BK clusters also included Ca$_V$1.3 channels, in SCG neurons as many as 10% of the BK cluster pixels included Ca$_V$1.3 channels, suggesting more intimate coupling of these interacting channels in SCG neurons. Randomized Ca$_V$1.3 channel clusters were distributed independently of the position of BK channels in the neurons (*Figure 5C,E*, orange circles). Such results show that endogenous BK and Ca$_V$1.3 channels in neurons exhibit a distribution and coupling similar to those in the expression system and suggest that clustering and proximity are fundamental properties of endogenous BK and Ca$_V$1.3 channels and not an artifact of overexpression.

Our original hypothesis was that low-voltage activated Ca$_V$1.3 channels can confer on BK channels the ability to be activated at quite negative voltages, and we have found that these channels are distributed in a spatial pattern consistent with functional coupling. Given that BK and Ca$_V$1.3 channels display a similar coordinated distribution in neurons, we tested whether BK channels are activated at −40 mV in these cells. We recorded potassium currents in dissociated hippocampal and sympathetic

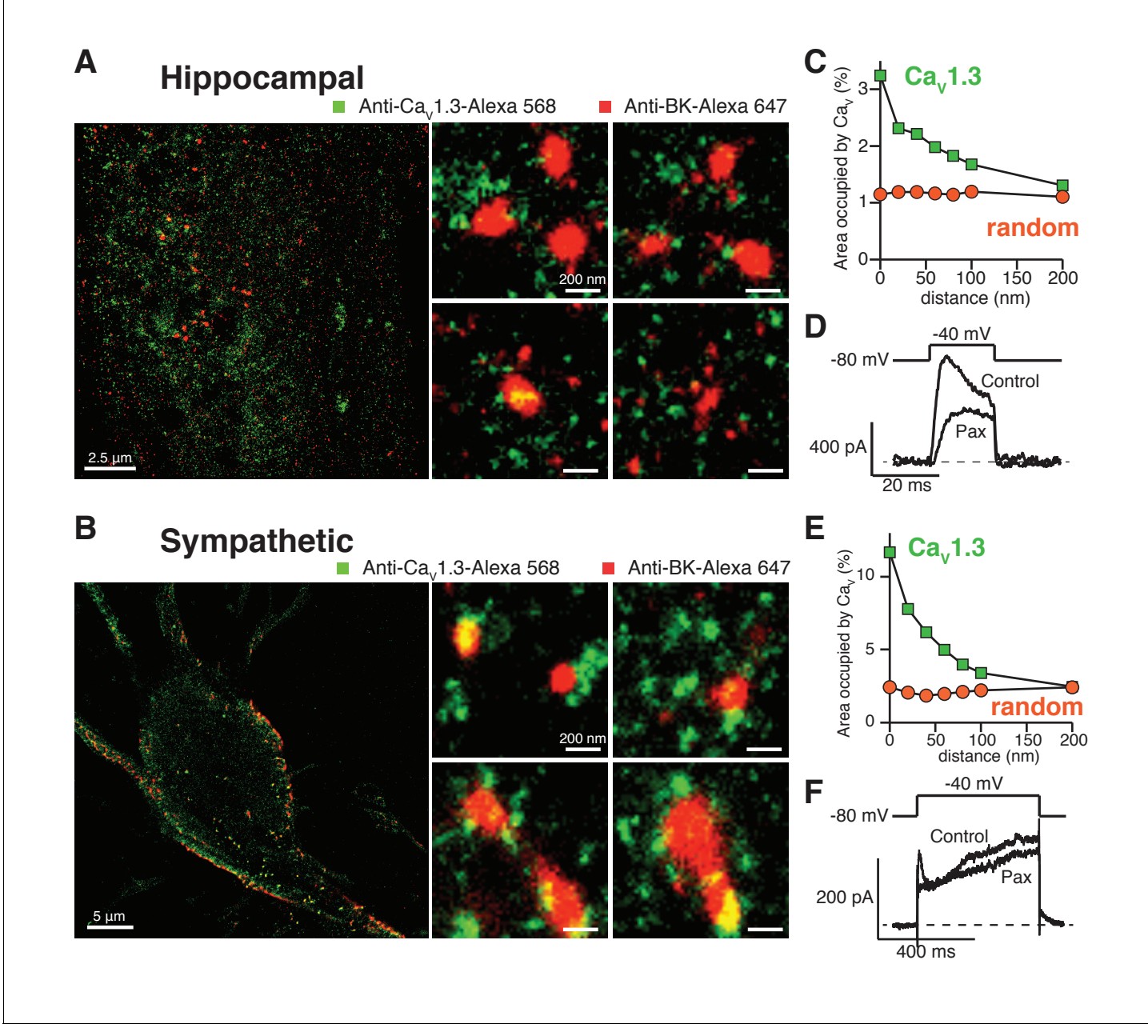

**Figure 5.** Endogenous BK and Ca$_V$1.3 channels also form clusters in neurons. (A, B) Representative single-molecule localization maps from cultured hippocampal (A) and sympathetic (B) neurons labeled for BK (red) and Ca$_V$1.3 (green) channels; zoom-in panels to the right show clusters of BK channels surrounded by Ca$_V$1.3 channels. (C, E) Distribution analysis based on the percentage of area occupied by Ca$_V$1.3 clusters (green squares) or randomized Ca$_V$1.3 clusters (orange circles, n = 6) as a function of the distance from BK channels in hippocampal (C, n = 6) and sympathetic (E, n = 7) neurons. (D, F) Outward currents activated at low negative potentials (−40 mV, see voltage protocol) before and after the application of 500 nM paxilline in hippocampal (D, n = 3) and sympathetic (F, n = 3) neurons.

The following figure supplements are available for figure 5:

**Figure supplement 1.** Specificity of anti-Ca$_V$1.3 antibody in tsA-201 cells expressing Ca$_V$1.3 and Ca$_V$1.2 channels.

**Figure supplement 2.** Clustering of BK and Ca$_V$1.3 channels is independent of the secondary antibody used.

**Figure supplement 3.** Single-channel activity of L-type calcium channels in sympathetic neurons at low voltage.

neurons before and after the application of 500 nM paxilline and analyzed subtracted currents for evidence of BK channel activity. *Figure 5D and F* show representative potassium currents activated at −40 mV. On average, we observed 200 ± 20 pA (n = 3) of paxilline-sensitive potassium currents at −40 mV in hippocampal neurons and 100 ± 20 pA (n = 3) in sympathetic neurons. Calcium current at −40 mV is minimal or undetectable in whole-cell recordings in hippocampal and sympathetic neurons (*Regan et al., 1991*; *Vivas et al., 2012*). Nonetheless, it is possible that activation of a few channels, or clusters, is enough to begin to activate BK channels at low voltages. To determine the activity of individual calcium channels at negative voltages, we recorded spontaneous calcium events in total internal reflection fluorescence (TIRF) microscopy. Sympathetic neurons were patched in whole-cell mode. The calcium indicator Rhod2 (200 μM) and the calcium chelator EGTA (10 mM) were included in the patch pipette to restrict calcium elevations to the plasma membrane. Membrane potential was held at −80 mV. Localized bursts of fluorescent events were detected (*Figure 5—figure supplement 3*). These events, termed sparklets, have been characterized in tsA-201 cells expressing $Ca_V1.3$ channels and are known to reflect the opening of single or clustered channels (*Navedo et al., 2007*). Importantly, these events increased in number and amplitude after application of 500 nM Bay K 8644 (*Figure 5—figure supplement 3D*), suggesting that they are mediated by L-type calcium channels. Altogether, these results show that SCG neurons exhibit spontaneous activity of single and coupled L-type calcium channels at negative potentials that can contribute to the activation of nearby BK channels even in the absence of detectable whole-cell calcium current.

## Discussion

We found that $Ca_V1.3$ channels couple to BK channels in multi-channel complexes, enabling BK channels to open at voltages as negative as −50 mV. Our single-molecule localization experiments did not support proposed models of strict stoichiometry such as a 1:1 ratio of $Ca_V$ and BK channels. Instead, we found that BK channels form clusters that are surrounded by clusters of $Ca_V1.3$ channels with no apparent fixed stoichiometry.

We provide several lines of evidence that support the hypothesis that these two channels congregate and are functionally coupled: (i) the functional and spatial coordination can be reconstituted in a heterologous system, (ii) the activation profile of BK channels changes with the expression of $Ca_V1.3$ channels, (iii) functional interaction persists when isolated in a patch pipette, (iv) coordinated gating is abolished by BAPTA, but not by EGTA in the cell, (v) the two channels co-immunoprecipitate, and (vi) they are physically close when observed under super-resolution microscopy. These findings are consistent with previous studies that show that BK channels co-precipitate with $Ca_V1.3$ channels from rat brain extracts (*Grunnet and Kaufmann, 2004*) and that chromaffin cells from mice lacking $Ca_V1.3$ channels exhibit smaller BK currents (*Marcantoni et al., 2010*).

Berkefeld and colleagues have previously shown that the activation profile of BK channels changes with the expression of different $Ca_V$ isoforms. For instance, in their work BK channels start activating at around −10 mV when co-expressed with $Ca_V2.1$ channels but start activating at around +10 mV when co-expressed with $Ca_V1.2$ channels (*Berkefeld et al., 2006*). In agreement, our results show that the BK channel activation curve corresponds closely to the activation of co-expressed $Ca_V2.2$ or $Ca_V1.3$ channels, respectively, and with the latter can start activating at voltages as negative as −50 mV. Our experiments differ, however, in that we did not co-express the auxiliary β subunit of BK channels, suggesting that this auxiliary subunit is not necessary to reconstitute the functional coupling between BK and $Ca_V$ channels. Our results add information about the composition of these complexes by using imaging at nanometer resolution, arguing against a 1:1 channel stoichiometry with a 10 nm distance between them. Instead, BK channels form homotypic clusters and then form higher order heterotypic clusters with clusters of $Ca_V$ isoforms.

Clustering is a characteristic of several ion channels, including potassium and calcium channels, (*Choi, 2014*; *King et al., 2014*; *Dixon et al., 2015*). Recently, it was shown that $Ca_V1.3$ channels form clusters that are functionally coupled to each other and gate cooperatively (*Moreno et al., 2016*). Here we show that BK channels arrange in clusters as well and that these clusters of BK channels arrange in higher order distributions surrounded by clusters of $Ca_V1.3$ channels. Multi-channel complexes between $Ca_V$ and BK channels have been proposed before. For example, in frog hair cells, it has been proposed that 85 calcium channels and 40 BK channels organize in the presynaptic active zone (*Roberts, 1994*). An important conclusion of that work is that, assuming the

simultaneous opening of 10 $Ca_V$ channels at any given time, the calcium concentration could reach >100 μM inside the active zone, decaying steeply at the edge. Others have proposed complexes with a ratio of 5:1 $Ca_V$ to BK channels (*Prakriya and Lingle, 2000*). In this case, calcium channels can be at distances as far as 50 nm from BK channels and would still be able to accumulate enough calcium to activate BK channels, if more than one calcium channel opened at the same time. The association we see between BK and $Ca_V1.3$ channels appears to have neither fixed stoichiometry nor fixed geometry. Rather it can be described as a statistical bias for proximity of clusters, a bias that would not be present with a random distribution. We describe the most common composition of this mega complex. Homotypic BK clusters have various sizes, with a median of 2000 $nm^2$ and appear to be nucleation centers for $Ca_V1.3$ clusters to come very close, forming a heterotypic cluster. By counting the number of $Ca_V1.3$ clusters within 50 nm from BK clusters, we estimated that there are on average 4 $Ca_V1.3$ clusters around a single BK cluster (n = 128 ROIs). However, the super-resolution data show considerable lack of symmetry, with some BK clusters lacking nearby $Ca_V1.3$ clusters, $Ca_V1.3$ clusters not touching BK clusters, and $Ca_V1.3$ clusters in close proximity to BK clusters at various distances and numbers. The mechanisms favoring the formation of higher order cluster between BK and $Ca_V1.3$ clusters might be (1) direct interactions between potassium and calcium channels, (2) indirect interactions mediated by scaffolding proteins, (3) membrane domains that restrict and concentrate their distribution, or (4) weak interactions between BK and $Ca_V1.3$ channels, that by lateral diffusion and mass action, result in aggregations around each other. The later alternative might explain why not all BK clusters are surrounded by $Ca_V1.3$ clusters, given an equilibrium where some clusters are in a non-interacting state while others are in an interacting state.

What concentration of intracellular calcium might be found in the center of this multi channel mega-complex? The amount of calcium reaching a BK channel cluster will depend on: (i) the number of $Ca_V1.3$ clusters around it (a mean of 4 according to our estimations), (ii) the number of $Ca_V1.3$ channels open at any given time, and (iii) their distance from the target BK channel. Interestingly, clustering of $Ca_V1.3$ channels leads to cooperative opening of $Ca_V1.3$ channels within the cluster (*Moreno et al., 2016*). It remains uncertain, however, whether a nearby second cluster of $Ca_V1.3$ channels could open at the same time as the first. Calcium entering from two equidistant $Ca_V1.3$ clusters around a BK cluster could sum linearly and therefore contribute equally to the activation of BK channels.

To conclude, we propose that the coupling of BK channels with low-voltage activated $Ca_V1.3$ channels in a multi-channel complex enables this feedback mechanism to activate BK channels at potentials near the threshold, preventing neurons from firing action potentials as we observed recently in sympathetic neurons (*Vivas et al., 2014*).

## Materials and methods

### Cell culture

As heterologous expression system we used tsA-201 cells (SIGMA, RRID:CVCL_2737). This cell line has been eradicated from mycoplasma at ECACC and its identity has been confirmed by STR profiling. They were grown in DMEM (Invitrogen) with 10% fetal bovine serum and 0.2% penicillin/streptomycin, passaged once a week, and incubated in 5% $CO_2$ at 37°C. Cells were transfected with 0.2–1 μg DNA per plasmid, plated for 24 hr after transfection, and used for experiments the next day. Lipofectamine 2000 (Invitrogen) was the chosen method of transfection. Successfully transfected cells were identified on the basis of green fluorescent protein (GFP) fluorescence.

Neurons were isolated from Sprague Dawley rats (RRID:RGD_5508397), which were handled according to guidelines approved by the University of Washington Institutional Animal Care and Use Committee. Neurons from superior cervical ganglion (SCG) were prepared from 7 to 12 week-old male rats by enzymatic digestion as described previously (*Vivas et al., 2013*). Isolated neurons were plated on poly-L-lysine (Sigma) coated glass chips and incubated in 5% $CO_2$ at 37°C in DMEM supplemented with 10% FBS and 0.2% penicillin/streptomycin.

Hippocampal neurons were prepared from newborn (P1) rats. The hippocampi of six pups were isolated and digested as described previously (*Moreno et al., 2016*). Neurons were plated on poly-D-lysine-coated coverslips at a density of $2 \times 10^5$ cells/coverslip and maintained with MEM

(Invitrogen) supplemented with 10% horse serum, 2% B27, 25 mM HEPES, 20 mM glucose, 2 mM GlutaMAX, 1 mM sodium pyruvate, and 1% penicillin/streptomycin. Neurons were incubated in 5% $CO_2$ at 35°C for two weeks before experiments and one-third of the medium was replaced every 5 days.

## Electrophysiological recordings

Calcium and potassium currents were recorded from tsA-201 cells and neurons using the whole-cell configuration with an EPC9 patch-clamp amplifier (HEKA). Patch pipettes had a resistance of 2–6 MΩ, and series resistances of 6–10 MΩ were compensated by 50–70%. Due to incomplete compensation, large currents at very depolarized voltages induced voltage errors >20 mV, artifactually reducing the steepness of recorded BK activation curves there. Other factors altering the activation of BK channels at different voltages are the inactivation of $Ca_V$, the calcium reversal potential, and the application of 0.1 mM BAPTA to the pipette solution. Liquid junction potentials were not corrected for. Currents were sampled at 10 KHz. The bath solution (Ringer's solution) contained 150 mM NaCl, 2.5 mM KCl, 2 mM $CaCl_2$, 1 mM $MgCl_2$, 10 mM HEPES, and 8 mM glucose, adjusted to pH 7.4 with NaOH. When recording from neurons, 100 nM TTX was added to block sodium channels. To isolate calcium currents, the pipette solution contained 140 mM CsCl, 20 mM TEA-Cl, 1 mM $MgCl_2$, 10 mM HEPES, 0.1 mM $Cs_4BAPTA$, 3 mM $Na_2ATP$, and 0.1 mM $Na_3GTP$, adjusted to pH 7.2 with CsOH. To confirm the identity of calcium currents, 100 µM $CdCl_2$ or 10 µM nifedipine was applied in the external solution. Potassium currents were recorded using a pipette solution containing 175 mM KCl, 1 mM $MgCl_2$, 5 mM HEPES, 0.1 mM $K_4BAPTA$, 3 mM $Na_2ATP$, and 0.1 mM $Na_3GTP$, adjusted to pH 7.2 with KOH. To confirm that potassium currents were from BK channels, 500 nM paxilline was applied in the external solution. Bath solution was superfused at 2 ml/min, permitting solution exchange surrounding the recording cell with a time constant of 4 s.

To determine the voltage dependence of activation gating (G-V curves), the peak current for inward calcium currents or the steady-state currents for outward potassium currents were divided by $(E - E_{rev})$, where E is the tested membrane potential and $E_{rev}$ is the reversal potential for calcium (+126 mV, assuming $[Ca^{2+}]_i$ = 100 nM) or potassium (−106 mV). These values were calculated using the Nernst equation. In *Figure 1*, the conductance G is normalized to Gmax and plotted as a function of the tested membrane potentials.

Single-channel currents were recorded in cell-attached configuration. The bath solution contained a high concentration of potassium to maintain the cytoplasmic potential close to 0 mV as well as a low free calcium. This solution contained 145 mM KCl, 2 mM $MgCl_2$, 2 mM $CaCl_2$, 2 mM EGTA, 10 mM HEPES, and 10 mM glucose, adjusted with KOH to pH 7.3. The pipette solution contained 2.5 mM KCl, 130 mM NaCl, 20 mM $CaCl_2$, 1 mM $MgCl_2$, 10 mM HEPES, and was adjusted to pH 7.3 with NaOH. Voltage steps to 0 mV were elicited from a holding potential of −80 mV for 1 s to activate $Ca_V$ channels. In some cases, 20 ms at the beginning of the traces was removed to correct for a transient artifact. Single-channel events were analyzed with pClamp 10.2 software.

## Co-immunoprecipitation and western blotting

To determine whether BK and $Ca_V1.3$ channels were interacting, we performed coimmunoprecipitation experiments. Cells transiently transfected with the channels were harvested in 500 µl of binding buffer (PBS containing 1 mM $NaVO_3$, 10 mM $Na^+$-pyrophosphate, 50 mM NaF, pH 7.4, and 1% Triton X-100), sonicated, and spun down at 30,000 g for 20 min. For the coimmunoprecipitation experiments, 100 µl of cell extract was incubated with 1 µg antibody overnight at 4°C. Then, 50 µl protein G Sepharose 4B beads (Thermo Fisher Scientific) was added to the mixture and incubated for an additional 4 hr. Beads were washed three times for 10 min with binding buffer. Proteins were released from the beads with 50 µl of SDS loading buffer. The protein mixture was loaded onto 8% tris-glycine SDS-PAGE gels and transferred onto polyvinylidene-fluoride membranes for Western-blot analysis.

## Single-molecule localization microscopy

Single-molecule localization microscopy was performed as previously described (*Moreno et al., 2016*). In brief, we used a ground-state depletion microscope system (SR-GSD, Leica) to generate single-molecule localization maps of $Ca_V1.3$ and BK channels in tsA-201 cells and neurons. The Leica

SR-GSD system was equipped with high-power lasers (488 nm, 1.4 kW/cm$^2$; 532 nm, 2.1 kW/cm$^2$; 642 nm, 2.1 kW/cm$^2$) and an additional 30 mW, 405 nm laser for backpumping. For all experiments, the camera was running in frame-transfer mode at a frame rate of 100 Hz (10 ms exposure time). To restrict the analysis of the fluorescence to the plane of the plasma membrane, the images were acquired in TIRF mode. 25,000–50,000 images were acquired and used to construct localization maps. We estimated a lateral localization accuracy of 16 nm for Alexa-647 (~1900 detected photons per switching cycle). Events with lower counts than 50 (Detection threshold) were treated as noise and discarded.

## Immunostaining

Cells were fixed with freshly prepared 3% paraformaldehyde and 0.1% glutaraldehyde for 15 min. After washing with PBS, aldehydes were reduced with 50 mM glycine for 10 min at 4°C, unspecific binding was blocked with 20% SEA BLOCK (Thermo Scientific), and permeabilized with 0.25% v/v Triton X-100 in PBS for 1 hr. Primary antibodies were used at 10 µg/ml in blocking solution and incubated overnight at 4°C. Secondary antibodies, labeled with Alexa-647 and Alexa-568 (Molecular Probes), at 2 µg/ml were incubated for 1 hr at room temperature. For single-molecule localization microscopy, fixed and stained cells were imaged in a buffer containing 10 mM $\beta$-mercaptoethylamine (MEA), 0.56 mg/ml glucose oxidase, 34 µg/ml catalase, and 10% w/v glucose in Tris-HCl buffer.

## Imaging Ca$_V$ sparklets with total internal reflection fluorescence (TIRF) microscopy

To detect calcium sparklets in SCG neurons we dialyzed an internal solution containing EGTA (10 mM) and the relatively fast Ca$^{2+}$ indicator, Rhod-2 (200 µM) to restrict Ca$^{2+}$ signals to the vicinity of the channels. We used a through-the-lens TIRF microscope built around an inverted microscope (IX-70; Olympus) equipped with a Plan-Apochromat (60X; NA 1.49) objective (Olympus) and an electron-multiplying charge-coupled device (EMCCD) camera (iXON; Andor Technology, UK). Images for the detection of sparklets were recorded at a frequency of 100 Hz using TILL Image software. Neurons were held to a membrane potential of −80 mV using the whole-cell configuration of the patch-clamp technique. Sparklets were detected and analyzed using custom software (Source code 2) written in MATLAB (RRID:SCR_001622). Fluorescence intensity values were converted to nanomolar units as described previously (*Navedo et al., 2007*).

## Plasmids and antibodies

DNA clones of Ca$_V$1.3 (#49333) and BK (#16195) channels were obtained from Addgene (RRID:SCR_002037). Auxiliary subunits for Ca$_V$1.3 channels, Ca$_V\beta$3 and Ca$_V\alpha$2$\delta$1 (from Diane Lipscombe, Brown University, RI), were transfected as well but no auxiliary subunits for BK channels. The plasmid of the BK channel tagged with GFP was kindly provided by James Trimmer (University of California Davis, CA). The plasmid of the Ca$_V$1.2 channel (kindly provided by Diane Lipscombe, Brown University, RI) was fused to the monomeric GFP variant GFP(A206K) (kindly provided by Eric Gouaux, Vollum Institute, OR). Ca$_V$1.3 channels were immunodetected using a rabbit primary antibody recognizing the residues 809 to 825 located at the intracellular II-III loop of the channel (DNKVTIDDYQEEAEDKD, kindly provided by Drs. William Catterall and Ruth Westenbroek) (*Hell et al., 1993*). The antibody against Ca$_V$2.2 recognizes residues 851–867 in an intracellular loop between domains II and III (Alomone Labs, RRID# AB_2039766). BK channels were detected using the anti-slo1 mouse monoclonal antibody clone L6/60 (Millipore Cat# MABN70 RRID:AB_10805948). Specificity of antibodies was tested in tsA-201 cells not transfected with either of the channels (*Figure 4—figure supplement 3*). The Ca$_V$1.2-EGFP was detected using rabbit anti-GFP conjugated to Alexa-647 (Molecular Probes Cat# A-31852 also A31852 RRID:AB_162553). The specificity of the Ca$_V$1.3 antibody against Ca$_V$1.2 channels was tested in tsA-201 cells transfected with both channels (*Figure 5—figure supplement 1*). α-Tubulin was detected using the rat monoclonal anti-tubulin clone YL1/2 (Abcam Cat# ab6160 RRID:AB_305328). The following Alexa-647 or Alexa-568 antibodies (Molecular Probes), were used: Donkey anti-rabbit 647 (Cat# A-31573 also A31573 RRID:AB_2536183), Donkey anti-rabbit 568 (Cat# A10042 RRID:AB_2534017), Donkey anti-mouse 647 (Cat# A-31571 also A31571 RRID:AB_

162542), Donkey anti-mouse 568 (Cat# A10037 RRID:AB_2534013), Chicken anti-rat 647 (Cat# A21472 RRID:AB_10375433), and Goat anti-rat 568 (Cat# A-11077 also A11077 RRID:AB_141874).

### Data analysis

Localization maps were reconstructed using LASAF software (Leica). Analysis of cluster size, distance between proteins, and area distribution was performed using binary masks in ImageJ software (NIH). Binary images were processed with close- (1 iteration and five counts) and watershed-macros before quantification.

We used IGOR Pro (IGOR Software, WaveMetrics, RRID:SCR_000325) and Excel (Microsoft) to analyze data. Data were collected from at least three independent experiments and are presented as Mean ± SEM. For the super-resolution cluster area data the median instead of the mean was used to describe the cluster size of BK and $Ca_V$ channels, given that the data were not normally distributed. A non-parametric statistical test (Kruskal-Wallis) was used to test for statistical significance between cluster sizes in different conditions. Other statistical analyses presented in this study were performed using parametric Student's t-test, considering p values <0.05 as statistical significance. The number of cells used for each experiment are detailed in each figure legend

## Acknowledgements

This study was supported by the National Institute of Health grants R37NS008174 (BH), the Wayne E Crill Endowed Professorship (BH), R01HL085686 (LFS), and R01HL085870 (LFS). We thank all members of the Hille laboratory and colleagues in the Department of Physiology and Biophysics at the University of Washington for discussions and experimental advice, and Lea M Miller for technical help. We thank Ximena Opitz-Araya and Andres Barria for providing hippocampal neuron cultures. We thank Jason Lambert for developing the macro in MATLAB to generate randomized single-molecule localization microscopy maps. We thank Ana de la Mata for her assistance with the Airyscan image acquisition. We thank Eammon J Dickson, Rose E Dixon, Duk-Su Koh, Martin Kruse, Manuel Navedo, Madeline Nieves, Jon T Sack, and Jie Zheng for their helpful comments on the manuscript.

## Additional information

### Funding

| Funder | Grant reference number | Author |
|--------|------------------------|--------|
| National Institute of Neurological Disorders and Stroke | R37NS008174 | Bertil Hille |
| National Heart, Lung, and Blood Institute | R01HL085686 | Luis F Santana |
| Wayne E. Crill Endowed Professorship | Professor Fellowship | Bertil Hille |
| National Heart, Lung, and Blood Institute | R01HL085870 | Luis F Santana |

The funders had no role in study design, data collection and interpretation, or the decision to submit the work for publication.

### Author contributions

OV, Conceptualization, Data curation, Formal analysis, Investigation, Methodology, Writing—original draft, Project administration, Writing—review and editing; CMM, Formal analysis, Investigation, Writing—original draft, Writing—review and editing; LFS, Funding acquisition, Methodology, Writing—review and editing; BH, Conceptualization, Supervision, Funding acquisition, Writing—original draft, Project administration, Writing—review and editing

### Author ORCIDs

Oscar Vivas, http://orcid.org/0000-0002-0964-385X
Luis F Santana, http://orcid.org/0000-0002-4297-8029

### Ethics

Animal experimentation: Animals were handled according to guidelines approved by the University of Washington Institutional Animal Care and Use Committee (#2084-03).

## Additional files

### Supplementary files

• Source code 1. Custom function to randomize binary particle images written in MATLAB.

• Source code 2. Custom software for $Ca^{2+}$ sparklet detection and analysis written in MATLAB.

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
