## [Decision Letter]

Thank you for submitting your article "Proximal clustering between BK and Ca_V_1.3 channels promotes functional coupling and BK channel activation at low voltage" for consideration by *eLife*. Your article has been favorably evaluated by Gary Westbrook (Senior Editor) and three reviewers, one of whom, Kenton J Swartz (Reviewer #1), is a member of our Board of Reviewing Editors. The following individual involved in review of your submission has agreed to reveal their identity: Christopher J Lingle (Reviewer #2).

The reviewers have discussed the reviews with one another and the Reviewing Editor has drafted this decision to help you prepare a revised submission.

Summary:

In this work by Vivas, Santana, Hille and colleagues the functional and spatial coupling of L-type calcium channels and BK channels were investigated. Through a series of very well done electrophysiology, molecular biology, and super-resolution imaging experiments, the nature of the interaction is shown. The authors specifically show 1) the mid-point of activation of the calcium channel influences BK channel activation, 2) membrane patches show dual activation, 3) channels co-immuno-precipitate, 3) EGTA and BAPTA differentially influence BK activation, and 4) GSD imaging shows clusters of BK channels surrounded by calcium channels in cells. Together these data present a model by which Ca_V_1.3 and BK channels co-assemble into clusters. The BK channels are an "island" surrounded by a variable number of Ca_V_1.3 channels that form a functional nano-domain on the membrane in model cells and neurons. In general, we think this paper is excellent. Much of the work is, in some way confirmatory of earlier work where L-type calcium channels and BK channels were shown to physically interact and to couple functionally. The unique (and new) data here, however, is the determination of the spatial organization of the channel clusters with fluorescence. The following are specific issues the authors should address in revision.

Essential revisions:

1) The authors use primary and secondary antibodies to label the channels for super-resolution imaging. The dimensions of this label complex (upwards of 30 nm) should be incorporated into any models where channel number or area are estimated.

2) Both Alexa-598 and Alexa-647 show significant spectral overlap. As far as we can tell no single-label controls were done to confirm that the two-color STORM images did not have any bleed-through or cross-talk. Because these images and analysis are so key to the paper, we would like to see a control where the labels are switched (Ca_V_1.3 is labeled with A647 and BK channels are labeled with A598 etc.) to confirm and control for the spatial distribution of these two channels and the relative organization of the channel complex.

3) It might also be nice to see a structured illumination (SIM) control of these complexes. This could be done with conventional FP probes in live cells. Thus, any influence of live cell vs. fixed cell could be addressed. The size of these clusters ~200-500nm should be resolvable with TIRF-SIM even in live cells.

4) There are details in the patch experiment that are missing, e.g., length of time at command voltage prior to illustrated traces, were paxilline and nifedipine applied in the same patch as the control and was it applied to the whole cell, whether by setting the cell in symmetrical K, the overall cytosolic [Ca^2+^] may reach a level for modest activation of BK unrelated to patch Ca^2+^ influx. The authors should also show unsubtracted K currents in Figure 1 so we can see exactly how the subtracted currents were obtained. The currents shown have some artifacts and it’s difficult to evaluate the quality of the voltage clamp. Also, please provide a description of the experimental and subtraction procedures.

5) Although it may not be easy to do, one can imagine other useful information contained in the images, e.g., comparisons of average BK channel cluster size. Also, in the analysis of Ca_V_1.3/BK data, shouldn't at farther distances (>200 nm) the presence of Ca_V_1.3 should fall to levels consistent with the "random" case? That would seem to be potentially important. If the distribution even out to 200 nm is still not totally random, that might support an interesting "mechanism" of coupling where lateral mobility of membrane proteins creates some associations simply from weak affinity interactions. The decided lack of symmetry in the Cav/BK clusters might also be more easily viewed within such a context.

6) The gel shown in Figure 2 is confusing. Based on earlier studies BK migrates at around 100 kDa, consistent with one of the bands shown. What is the band at 250 kDa? Also, was this experiment done more than once?

---

## [Author Response]

*Essential revisions:*

*1) The authors use primary and secondary antibodies to label the channels for super-resolution imaging. The dimensions of this label complex (upwards of 30 nm) should be incorporated into any models where channel number or area are estimated.*

The reviewer raises a particularly important point regarding the analysis of super-resolution images. We agree that the labeling of proteins with antibodies could lead to an over estimation of the area of protein clusters. Thus, determining the number of channels within each cluster using GSD imaging is difficult. Accordingly, we decided to discuss the composition of the clusters only in terms of area without reaching any conclusion regarding the number of channels within the cluster. We deleted a sentence in the Results section that said “In each pixel with dimensions of 20 by 20 nm, two channels could fit”. We also deleted the sentences in the Discussion section that described the number of BK and Ca_V_1.3 channels per cluster. Finally, we removed the schematic model in Figure 6.

*2) Both Alexa-598 and Alexa-647 show significant spectral overlap. As far as we can tell no single-label controls were done to confirm that the two-color STORM images did not have any bleed-through or cross-talk.*

We agree with the reviewers that controls for bleed-through were missing in the previous version of the manuscript. We now provide new data addressing this point. Figure 4—figure supplement 2 shows immunostaining of microtubules using the two different secondary antibodies and calculating the fraction of crosstalk. The fraction of crosstalk of the emission of Alexa-568 into the 647 channel is only 0.092 ± 0.032. The fraction of crosstalk of the emission of Alexa-647 into the 568 channel is only 0.008 ± 0.001. We also give details of how these experiments were conducted in the second paragraph of the Results subsection “Clusters of Ca_V_1.3 channels surround BK channels”. It is important to note that we used Alexa-568 and not Alexa-598. We have corrected any typos related to this throughout the manuscript.

*Because these images and analysis are so key to the paper, we would like to see a control where the labels are switched (Ca_V_1.3 is labeled with A647 and BK channels are labeled with A598 etc.) to confirm and control for the spatial distribution of these two channels and the relative organization of the channel complex.*

We agree with the reviewers that these controls were necessary. Although we had failed to clarify this in the first version of the manuscript, we in fact had performed these experiments using reciprocal immunostaining using Alexa-647 for the Ca_V_1.3 and Alexa-568 for the BK channels in SCG neurons. Giving that BK and Ca_V_1.3 clusters and the BK-Ca_V_1.3 organization was independent of the secondary antibody used, the previous version of the analysis shown in Figure 5 already included data from both immunostainings. In order to clarify this we now added Figure 5—figure supplement 2 to show representative super-resolution maps of the reciprocal immunostaining and added a section to the Results explaining this figure:

“Reciprocal immunostaining using Alexa-647 for the Ca_V_1.3 and Alexa-568 for the BK channels was also performed. The BK-Ca_V_1.3 clustering organization was not dependent on the secondary antibody used (Figure 5—figure supplement 2). Independent of the secondary antibody used, all the cells were included in the distribution analysis”.

*3) It might also be nice to see a structured illumination (SIM) control of these complexes. This could be done with conventional FP probes in live cells. Thus, any influence of live cell vs. fixed cell could be addressed. The size of these clusters ~200-500nm should be resolvable with TIRF-SIM even in live cells.*

We thank the reviewers for suggesting these experiments in living cells, as they help us rule out the possibility that the clusters we observed are an artifact of fixation. We performed experiments expressing BK-GFP and Ca_V_1.3-Ruby tagged channels in tsA-201 cells and imaged the cells in the focal plane of the membrane using a ZEISS AiryScan microscope, which has a lateral resolution of about 140 nm. The AiryScan resolution is similar to that of SIM microscopy (about 100 nm). New Figure 4—figure supplement 4 shows representative cells in which big clusters of BK channels are surrounded by Ca_V_1.3 clusters in agreement with our previous observations in fixed cells. We added a section describing this result:

“To rule out that this distribution was a fixation artifact, we imaged the clusters from live tsA-201 cells in an AiryScan microscope which doubles the resolution achieved with a diffraction-limited confocal microscope. For these experiments we expressed BK channels tagged with GFP and Ca_V_1.3 channels tagged with mRuby. Similar to what we observed in fixed cells, live cells exhibited BK channel puncta surrounded by Ca_V_1.3 channel puncta (Figure 4—figure supplement 4)”.

4) There are details in the patch experiment that are missing, e.g., length of time at command voltage prior to illustrated traces.

We agree with the reviewers and are providing more details in our patch-clamp experiments. In some traces, only 20 ms at the beginning was removed to correct for a transient artifact. We added a sentence in the Materials and methods section (subsection “Electrophysiological recordings”, last paragraph) and added Figure 2—figure supplement 1 to show a whole representative trace. We also noticed that there was a typo in the scale bar in Figure 2. It should be 20 ms, not 200 ms. We corrected this error.

Were paxilline and nifedipine applied in the same patch as the control and was it applied to the whole cell?

Yes, nifedipine and paxilline were applied sequentially to the patches after recording in control conditions. Yes, it was applied to the whole cell. This has been clarified in legend of Figure 2 and Results (subsection “Ca_V_1.3 channels are in close proximity to BK channels”, first paragraph).

*Whether by setting the cell in symmetrical K, the overall cytosolic [Ca^2+^] may reach a level for modest activation of BK unrelated to patch Ca^2+^ influx.*

This is an important point and we thank the reviewers for raising it. In the Materials and methods section (subsection “Electrophysiological recordings”, last paragraph), we stated that we used a bath solution with low free calcium by adding 2 mM EGTA, which results in 10 μM free calcium. This amount of calcium outside the cell would decrease the driving force by half, given that Erev for calcium would go from 127 mV to 64 mV. Activation of Ca_V_1.3 channels by depolarizing the cell to 0 mV (with a symmetrical K solution) would lead to calcium influx. The amount of calcium entering the cell will be reduced by endogenous buffers. In some mammalian cells ^1,2^, the buffer strength is 99%, suggesting that from 10 μM outside the cell, less than 100 nM would enter the cell. The inactivation of Ca_V_1.3 channels and the activation of extrusion mechanisms would also reduce calcium influx. For these reasons, we think that the contribution of calcium influx unrelated to the activation of Ca_V_1.3 channels in the patch is minimal. We clarified in the Results section this point: “To prevent the activation of BK channels inside the patch by calcium coming from Ca_V_1.3 channels outside the patch, 2 mM EGTA was added to the bath solution”.

1) Neher E, Augustine GJ. 1992. Calcium gradients and buffers in bovine chromaffin cells. J Physiol. 450:273-301.

2) Tse A, Tse FW, Hille B. 1994. Calcium homeostasis in identified rat gonadotrophs. J Physiol. 477 (Pt 3):511-25.

The authors should also show unsubtracted K currents in Figure 1 so we can see exactly how the subtracted currents were obtained. The currents shown have some artifacts and it’s difficult to evaluate the quality of the voltage clamp.

We added unsubtracted traces to Figure 1 and rearranged the figure. It also helps to illustrate how the subtraction was done. Thank you for the suggestion.

*5) Although it may not be easy to do, one can imagine other useful information contained in the images, e.g., comparisons of average BK channel cluster size.*

We appreciate that the reviewers elaborated on the mechanism of coupling between these channels. We performed the suggested analysis. The median cluster size of BK channels is 1600 nm^2[2]^ when co-expressed with Ca_V_1.3 channels (n = 5 cells) and 2000 nm^2[2]^ when co-expressed with Ca_V_2.2 channels (n = 4 cells). This is not significantly different. We provided the data in the third and sixth paragraphs of the subsection “Clusters of Ca_V_1.3 channels surround BK channels”.

*Also, in the analysis of Ca_V_1.3/BK data, shouldn't at farther distances (>200 nm) the presence of Ca_V_1.3 should fall to levels consistent with the "random" case? That would seem to be potentially important. If the distribution even out to 200 nm is still not totally random, that might support an interesting "mechanism" of coupling where lateral mobility of membrane proteins creates some associations simply from weak affinity interactions. The decided lack of symmetry in the Cav/BK clusters might also be more easily viewed within such a context.*

The reviewers are correct, given the nature of the analysis, the density of Ca_V_1.3 channels must fall to levels consistent with the random case after 200 nm. It is important to note that the density of Ca_V_1.3 channels does fall to random levels in SCG and hippocampal neurons (Figure 5) supporting such prediction of the analysis. To be able to analyze the presence of Ca_V_1.3 clusters at distances further away than 200 nm, an experiment with very low expression of BK channels would be ideal. We expanded on the coupling by weak interactions in the Discussion section:

“4) weak interactions between BK and Ca_V_1.3 channels, that by lateral diffusion and mass action, result in aggregations around each other. The later alternative might explain why not all BK clusters are surrounded by Ca_V_1.3 clusters, given an equilibrium where some clusters are in a non-interacting state while others are in an interacting state”.

*6) The gel shown in Figure 2 is confusing. Based on earlier studies BK migrates at around 100 kDa, consistent with one of the bands shown. What is the band at 250 kDa? Also, was this experiment done more than once?*

We agree with the reviewers that this point needed clarification. The molecular weight for the BK channel has been reported to be between 100 to 130 KDa using the clone L6/60 of the anti-Slo1 antibody. The second band observed around 250 KDa likely corresponds to a BK dimer, as it has been seen and proposed by other groups^3,4^. We have clarified this in the legend of Figure 2.

3) Ridgway LD, Kim EY, Dryer SE. 2009. MAGI-1 interacts with Slo1 channel proteins and suppresses Slo1 expression on the cell surface. American Journal of Physiology. 297(1): C55-C65. DOI: 10.1152/ajpcell.00073.2009.

4) Yan J, Aldrich RW. 2010. LRRC26 auxiliary protein allows BK channel activation at resting voltage without calcium. Nature. 466(7305):513-6. DOI: 10.1038/nature09162